# Retrospective cohort study evaluating clinical, biochemical and pharmacological prognostic factors for prostate cancer progression using primary care data

Samuel William David Merriel  ,[1] Suzanne Marie Ingle,[2] Margaret T May,[2,3] Richard M Martin[2,3]

¹College of Medicine & Health, University of Exeter, Exeter, UK
²Bristol Medical School, Population Health Sciences, University of Bristol, Bristol, UK
³National Institute for Health Research (NIHR) Biomedical Research Centre, University Hospitals Bristol NHS Foundation Trust and the University of Bristol, Bristol, UK

**Correspondence to**
Dr Samuel William David Merriel;
s.w.d.merriel@exeter.ac.uk

## ABSTRACT

**Objectives** To confirm the association of previously reported prognostic factors with future progression of localised prostate cancer using primary care data and identify new potential prognostic factors for further assessment in prognostic model development and validation.

**Design** Retrospective cohort study, employing Cox proportional hazards regression controlling for age, prostate specific antigen (PSA), and Gleason score, was stratified by diagnostic stage.

**Setting** Primary care in England.

**Participants** Males with localised prostate cancer diagnosedbetween 01/01/1987 and 31/12/2016 within the Clinical Practice ResearchDatalink database, with linked data from the National Cancer Registration andAnalysis Service and Office for National Statistics.

**Primary and secondary outcomes** Primary outcome measure was prostate cancer mortality. Secondary outcome measures were all-cause mortality and commencing systemic therapy. Up-staging after diagnosis was not used as a secondary outcome owing to significant missing data.

**Results** 10 901 men (mean age 74.38±9.03 years) with localised prostate cancer were followed up for a mean of 14.12 (±6.36) years. 2331 (21.38%) men underwent systemic therapy and 3450 (31.65%) died, including 1250 (11.47%) from prostate cancer. Factors associated with an increased risk of prostate cancer mortality included age; high PSA; current or ex-smoker; ischaemic heart disease; high C reactive protein; high ferritin; low haemoglobin; high blood glucose and low albumin.

**Conclusions** This study identified several new potential prognostic factors for prostate cancer progression, as well as confirming some known prognostic factors, in an independent primary care data set. Further research is needed to develop and validate a prognostic model for prostate cancer progression.

## INTRODUCTION

Prostate cancer prognosis and treatment decisions remain a challenging clinical area for clinicians and patients, particularly for men

## Strengths and limitations of this study

► Large retrospective cohort study of men with localised prostate cancer.
► Mean follow-up of 14.12 years.
► Data available on a wide range of potential prognostic factors for prostate cancer progression.
► Missing cancer stage and grade data from National Cancer Research and Analysis Service cancer registry excluded a proportion of the cohort.

with localised disease at the time of diagnosis. In recent decades, prostate cancer detection rates in many countries have increased markedly, in part, as a result of the rising use of asymptomatic prostate specific antigen (PSA) testing;[1] however, more intensive PSA-based detection of prostate cancer has not been convincingly directly correlated with reductions in prostate cancer mortality for all men,[2] implying increasing overdetection of clinically insignificant tumours.[3] Treatments for prostate cancer carry a significant risk of morbidity for men,[4 5] underlining the importance of being able to identify which men with tumours confined to the prostate at diagnosis are at higher risk of prostate cancer progression and mortality to inform discussions about management options.

Defining and measuring cancer progression with respect to treatment studies is outlined in the Response Evaluation Criteria in Solid Tumours (RECIST) criteria, which was originally published by the World Health Organisation (WHO) in 2000[6] and most recently updated in 2009.[7] Evidence of tumour shrinkage on imaging and time to development of disease progression are used to measure treatment response. Definitions of cancer progression that are relevant to

prognostic studies are less well defined, and numerous clinical, biological and surrogate markers of progression have been proposed in various studies. Prostate cancer mortality appears to be the logical ultimate endpoint of prostate cancer progression, but other measures such as development of metastases,[8] biochemical recurrence,[9] commencing systemic therapy[10] and protein expression[11] have also been reported.

There are a plethora of prognostic factor studies and prediction tools for prostate cancer risk[12] and prognosis[13] in the published literature. The vast majority are not externally calibrated or validated, and very few are established for use in clinical practice.[12] Initiatives such as the MRC PROGnosis RESearch Strategy Partnership (PROGRESS) partnership highlight the importance of high-quality prognostic research to help inform clinical practice[14] and outline methodologically rigorous approaches to achieve this aim.[15–17] Developing clinically useful risk-prediction rules starts with identifying potentially important prognostic factors, which could be incorporated into a prediction model. The aim of the current study is to confirm the association of previously reported prognostic factors with future progression of localised prostate cancer using primary care data and identify new potential prognostic factors for further assessment in prognostic model development and validation.

## MATERIALS AND METHODS

The protocol for this study has been published previously in *BMJ Open*.[18] In summary, we undertook a retrospective cohort study using a longitudinal data set of prospectively collected electronic primary care medical records from general practices (GPs) in England for the Clinical Practice Research Datalink (CPRD).[19] This data set was linked with cancer registry data from the National Cancer Research and Analysis Service (NCRAS)[20] and mortality data from the Office for National Statistics (ONS).[21] Men were included if they had a diagnosis of prostate cancer entered into their medical record during the 20-year study period (01 January 1987—31 December 2016). Localised prostate cancer was defined as T1-2/N0/M0 based on staging data entered into the NCRAS registry, which is determined from a combination of clinical, pathological and radiological data.[22]

Potentially relevant clinical, biochemical and pharmacological factors measured in CPRD were identified from a review of the existing published literature (See *BMJ Open* protocol paper[18] for more information about the prognostic factors assessed). The primary outcome of the study was prostate cancer mortality. Secondary outcomes were all-cause mortality and commencing systemic prostate cancer therapy (a measurable proxy for progression and metastatic spread of prostate cancer). Surgery, radiotherapy and brachytherapy were classified as localised therapy, with chemotherapy, hormone treatments (primary or neoadjuvant) and immunotherapy considered systemic therapy. Mortality outcomes were based

on primary/immediate cause of death reported in death certification information from the ONS and therapy outcomes from NCRAS data. In our published protocol,[18] up-staging after diagnosis was proposed as a secondary outcome indicating spread of disease; however, this was not used in the final analysis as repeat staging was rarely recorded in the cancer registry.

Descriptive statistics were used to summarise the basic demographic details of the men and the prevalence of the preselected putative prognostic factors. Cox proportional hazards regression was used to estimate crude and mutually adjusted hazard ratios (with 95% CIs) for prostate cancer-specific and all-cause mortality according to the prognostic factors, controlling for variables currently used in clinical practice (age, PSA level, Gleason score). Regression analyses of continuous prognostic factors were standardised using hazard ratios per change in one SD. A proportional hazards test was performed to confirm modelling met regression assumptions. The analysis was also stratified by stage at diagnosis (T1/2N0M0 vs T3+and/or N1 and/or M1). Sensitivity analysis was performed, assuming all men in the overall sample with unknown tumour location had localised disease. In order to achieve 95% power and detect a difference of 0.1 in prostate cancer mortality for a binary risk factor using an alpha of 0.05, a sample of at least 6046 men with prostate cancer would be required, assuming that 10% die over a median 10-year follow-up.

## RESULTS

A total of 54 500 men within CPRD had a diagnosis of prostate cancer entered into their primary care medical record during the study period. Baseline participant data are shown in table 1. Tumour–node–metastases (TNM) staging data from the linked cancer registry were available for 7646 (14.03%) of the sample population and treatment data were available for 22 766 (41.77%) men. Missing TNM staging data from the cancer registry were lower for men diagnosed in more recent years: there were no TNM stage data for men diagnosed before 1993, rising to 37.2% with TNM stage data (1064/2836) in 2015. This is consistent with a recent validation study of the NCRAS prostate cancer registry that showed low levels of completeness of TNM stage and Gleason score data prior to 2010.[23] Using the available staging and treatment data, 10 901 (20%) men were identified as having localised prostate cancer at the time of diagnosis and were included in the final cohort for analysis, with a mean follow-up of 14.12 (±6.36) years. Levels of missing data for selected prognostic factors within CPRD varied.

1250 men with localised disease died of prostate cancer over the course of follow-up, giving a prostate cancer mortality rate of 8.1 per 1000 person-years. The total number of deaths for included men was 3450 (21.11 deaths per 1000 person-years). A total of 2331 (21.38%) men with localised disease received systemic therapy in the follow-up period after diagnosis. For over 90% of the

**Table 1**  Baseline participant data

| | Localised n=10 901 | Missing data |
|---|---|---|
| Mean (SD) | | |
| Age (years) | 74.38 (±9.03) | 0% |
| BMI (kg/m$^2$) | 27.43 (±4.48) | 5.64% |
| Follow-up (years) | 14.12 (±6.36) | 0% |
| Median (IQR) | | |
| PSA (ng/mL) | 8.4 (5.55, 14.6) | 30.66% |
| n (%) | | |
| Gleason score | | |
| 6 | 3655 (33.53%) | 33.23% |
| 7+ | 4420 (40.55%) | |
| Family history of prostate cancer | 70 (0.64%) | 55.11% |
| Ethnicity | | |
| White | 7361 (67.53%) | 29.79% |
| Mixed | 21 (0.19%) | |
| Asian | 75 (0.69%) | |
| Black | 156 (1.43%) | |
| Other | 41 (0.38%) | |

BMI, body mass index; PSA, prostate specific antigen.

men, it was unknown whether they were reinvestigated for cancer staging after diagnosis or not (see table 2).

Raised acute phase reactants (C reactive protein (CRP) (adjusted HR per SD 1.35 95% CI 1.02 to 1.77)), ferritin (adjusted HR per SD 2.03; 95% CI 1.21 to 3.39) and random glucose (adjusted HR per SD 1.27; 95% CI 1.06 to 1.54) were associated with prostate cancer mortality. Anaemia (adjusted HR per SD 0.72; 95% CI 0.59 to 0.88) and low albumin (adjusted HR per SD 0.81; 95% CI 0.67 to 0.97) were also associated with this outcome. No medications assessed were associated with prostate cancer mortality. Current and ex-smokers (adjusted HR 1.47; 95% CI 1.05 to 2.05) and patients with a history of ischaemic heart disease (adjusted HR 1.79; 95% CI 1.20 to

2.66) had a higher risk of prostate cancer mortality over the study period.

Raised CRP, anaemia and low albumin were biochemical factors associated with all-cause mortality; with anaemia and low albumin also being associated with commencing systemic therapy. A number of other factors were also associated with all-cause mortality, including age, raised PSA, smoking and smoking-related disease, cardiovascular diseases as well as current use of aspirin or beta-blockers. Smoking and beta-blockers were also associated with increased risk of systemic therapy, as were vitamin D supplements. Benign prostatic hyperplasia and alpha-blocker prescription were associated with a reduced risk of commencing systemic therapy (see tables 3 and 4 for adjusted analysis results and online supplemental tables S1 and S2 for unadjusted results).

Sensitivity analysis including all participants with unknown tumour location showed a relationship between a history of stroke and all-cause mortality (adjusted HR 1.47; 95% CI 1.12 to 1.93 p=0.006). The relationship between aspirin and prostate cancer mortality altered to very weak evidence for association (adjusted HR 1.55 95% CI 0.79 to 3.02 p=0.2). For all other factors measured and for all three outcomes in the analysis, the direction of relationship did not change and the magnitude of relationship stayed relatively stable (see online supplemental tables S3–6).

## DISCUSSION

This retrospective cohort study used primary care medical records data for men with localised prostate cancer from CPRD to confirm prognostic factors associated with prostate cancer progression. Well-known factors already incorporated into clinical guidelines, such as age and PSA, were confirmed as being individual prognostic factors. In addition, further clinical (history of smoking or ischaemic heart disease) and biochemical (anaemia or high ferritin) factors were found to be strongly associated with prostate cancer mortality. Anaemia, low albumin, raised PSA, history of ischaemic heart disease and smoking were also strongly associated with all-cause mortality, as were

**Table 2**  Primary and secondary outcomes for included and excluded participants

| | | Prostate cancer mortality | All-cause mortality | Systemic therapy | Upstaging* |
|---|---|---|---|---|---|
| Included | Localised (T1/2 N0 M0) n=10 901 | 1250 (11.47%) | 3450 (31.65%) | 2331 (21.38%) | 45 (0.41%) |
| Excluded | Invasive (T3+/N1/M1) n=12 318 | 3894 (31.61%) | 6916 (56.15%) | 10 881 (88.33%) | 28 (0.23%) |
| | Unknown n=31 281 | 1540 (4.92%) | 5420 (17.33%) | 31 954 (58.63%) | 19 (0.06%) |

*Repeat staging data missing for 50 119 (91.96%) of sample.

**Table 3** Prognostic factors for men with localised disease associated with outcomes

n=10901

| Factor | Mean (SD) | Missing (n (%)) | Prostate cancer mortality | | | All-cause mortality | | | Systemic therapy | | |
|---|---|---|---|---|---|---|---|---|---|---|---|
| | | | HR per SD** | 95% CI | P | HR per SD** | 95% CI | P | HR per SD** | 95% CI | P |
| Age | 74.39 (9.03) | 0 | 1.70 | 1.40 to 2.06 | <0.01 | 1.92 | 1.74 to 2.12 | <0.01 | 1 | 0.95 to 1.06 | 0.98 |
| BMI | 27.43 (4.48) | 394 (3.61%) | 1.05 | 0.90 to 1.08 | 0.52 | 0.97 | 0.90 to 1.05 | 0.51 | 1.04 | 0.99 to 1.09 | 0.10 |
| Triglycerides | 1.45 (0.80) | 3856 (35.37%) | 0.83 | 0.64 to 1.08 | 0.16 | 1.00† | 0.90 to 1.13 | 0.93 | 1.03 | 0.97 to 1.09 | 0.37 |
| HDL cholesterol | 1.35 (0.43) | 3954 (36.27%) | 1.05 | 0.89 to 1.23 | 0.56 | 1.01† | 0.91 to 1.12 | 0.86 | 1.01 | 0.95 to 1.07 | 0.75 |
| LDL cholesterol | 2.95 (0.99) | 4698 (43.10%) | 0.86 | 0.69 to 1.07 | 0.18 | 0.92† | 0.82 to 1.02 | 0.12 | 0.99 | 0.94 to 1.05 | 0.86 |
| Hb | 144.28 (14.35) | 2696 (24.73%) | 0.72 | 0.59 to 0.88 | <0.01 | 0.74 | 0.67 to 0.82 | <0.01 | 0.92 | 0.86 to 0.98 | 0.01 |
| Albumin | 41.83 (3.94) | 2954 (27.10%) | 0.81 | 0.67 to 0.97 | 0.02 | 0.83 | 0.76 to 0.91 | <0.01 | 0.94 | 0.89 to 0.99 | 0.04 |
| Random glucose | 5.70 (2.11) | 4525 (41.51%) | 1.27 | 1.06 to 1.54 | 0.01 | 1.12 | 0.99 to 1.25 | 0.06 | 1.02† | 0.95 to 1.09 | 0.66 |
| | Median (IQR) | Missing (n (%)) | | | | | | | | | |
| PSA | 8.4 (5.55, 14.60) | 2352 (21.58%) | 1.71 | 1.32 to 2.23 | <0.01 | 1.46 | 1.19 to 1.78 | <0.01 | 1.34 | 1.06 to 1.68 | 0.01 |
| CRP | 3.9 (2, 8) | 8061 (73.95%) | 1.35† | 1.02 to 1.77 | 0.03 | 1.23† | 1.05 to 1.45 | 0.01 | 1.07 | 0.95 to 1.20 | 0.24 |
| Ferritin | 108.6 (47, 196) | 9495 (87.10%) | 2.03 | 1.21 to 3.39 | <0.01 | 0.98† | 0.60 to 1.59 | 0.93 | 1.05 | 0.85 to 1.31 | 0.64 |

*Adjusted for age, PSA, Gleason score, TNM stage.

†Proportional Hazards assumption test not met.

BMI, body mass index; CRP, C reactive protein; Hb, haemoglobin; HbA1c, haemoglobin A1c; HDL, high density lipoprotein; LDL, low density lipoprotein; PSA, prostate specific antigen; TNM, tumour–node–metastases.

**Table 4** Prognostic factors for men with localised disease associated with outcomes

**n=10901**

| Factor | n (%) | Missing (n(%)) | Prostate cancer mortality | | | All-cause mortality | | | Systemic therapy | | |
|---|---|---|---|---|---|---|---|---|---|---|---|
| | | | HR* | 95% CI | P | HR* | 95% CI | P | HR* | 95% CI | P |
| Smoker (current/ ex-) | 5112 (46.89%) | 777 (7.13%) | 1.47 | 1.05 to 2.05 | 0.02 | 1.66 | 1.39 to 1.98 | <0.01 | 1.21 | 1.09 to 1.33 | <0.01 |
| Excess alcohol | 1829 (16.78%) | 4370 (40.09%) | 0.61 | 0.36 to 1.04 | 0.07 | 0.91† | 0.71 to 1.18 | 0.47 | 0.99 | 0.87 to 1.13 | 0.88 |
| BPH | 1169 (10.72%) | 3526 (32.35%) | 0.64 | 0.36 to 1.11 | 0.11 | 0.81 | 0.62 to 1.05 | 0.11 | 0.76 | 0.65 to 0.90 | <0.01 |
| COPD | 862 (7.91%) | 3583 (32.87%) | 0.86 | 0.47 to 1.57 | 0.63 | 1.64 | 1.29 to 2.09 | <0.01 | 1.18 | 0.99 to 1.41 | 0.06 |
| CVA | 553 (5.07%) | 3584 (32.88%) | 0.90 | 0.42 to 1 to94 | 0.79 | 1.19 | 0.85 to 1.68 | 0.30 | 0.92 | 0.72 to 1.17 | 0.49 |
| IHD | 1548 (14.20%) | 3405 (31.24%) | 1.79 | 1.20 to 2.66 | <0.01 | 1.25 | 1.02 to 1.55 | 0.04 | 1.01 | 0.87 to 1.18 | 0.86 |
| PVD | 202 (1.85%) | 3582 (32.86%) | 2.24 | 0.98 to 5.12 | 0.06 | 1.91 | 1.24 to 2.95 | <0.01 | 1.04 | 0.71 to 1.51 | 0.85 |
| T2DM | 1508 (13.83%) | 3448 (31.63%) | 0.97 | 0.62 to 1.51 | 0.89 | 0.95 | 0.76 to 1.19 | 0.68 | 0.99 | 0.86 to 1.14 | 0.91 |
| Aspirin | 426 (3.91%) | 16 (0.15%) | 1.88 | 0.96 to 3.70 | 0.06 | 1.58 | 1.09 to 2.29 | 0.02 | 1.24 | 0.95 to 1.60 | 0.11 |
| Metformin | 33 (0.30%) | | | | | 2.74 | 0.88 to 8.49 | 0.08 | 1.73 | 0.77 to 3.85 | 0.18 |
| Alpha-blockers | 305 (2.80%) | | 1.55 | 0.72 to 3.35 | 0.26 | 1.15 | 0.76 to 1.73 | 0.52 | 0.57 | 0.39 to 0.82 | <0.01 |
| Beta-blockers | 265 (2.43%) | | 2.03 | 0.89 to 4.60 | 0.09 | 1.79 | 1.18 to 2.72 | <0.01 | 1.48 | 1.09 to 1.99 | 0.01 |
| Statins | 339 (3.11%) | | 1.65 | 0.87 to 3.15 | 0.13 | 1.01 | 0.66 to 1.53 | 0.97 | 1.06 | 0.84 to 1.34 | 0.61 |
| Vitamin D | 465 (4.27%) | | 1.33 | 0.65 to 2.71 | 0.44 | 1.13 | 0.78 to 1.65 | 0.51 | 1.35 | 1.09 to 1.68 | <0.01 |

*Adjusted for age, PSA, Gleason score, TNM stage.
†Proportional Hazards assumption test not met.
BPH, Benign Prostatic Hypertrophy; COPD, chronic obstructive pulmonary disease; CVA, cerebrovascular accident; IHD, ischaemic heart disease; PVD, peripheral vascular disease; T2DM, type 2 diabetes mellitus.

peripheral vascular disease, chronic obstructive pulmonary disease and beta-blocker use. Smoking history was strongly associated with future systemic therapy, as were recent prescriptions of alpha-blockers or vitamin D supplements.

This analysis confirms the prognostic associations of some factors in prostate cancer progression. Smoking has also been found to be a risk factor for prostate cancer progression and mortality in cohort studies[24] and systematic reviews.[8] Low albumin was associated with prostate cancer mortality in the Apolipoprotein-related MOrtality RISk (AMORIS) cohort[25] and, along with anaemia,[26] was a more widely accepted predictor of poor cancer outcomes.[27] The published literature around the prognostic effect of beta-blockers for prostate cancer patients has been more mixed,[28] with this study lending weight to the evidence of increased mortality in patients with cancer. Body mass index (BMI) was not shown to be associated with prostate cancer and overall mortality in this study. While some observational studies of prostate cancer have suggested that an association may exist,[8 29 30] reviews of trial data have demonstrated that higher BMI may actually improve the prognosis for men with cancer.[31]

This study attempted to confirm prognostic factors in a primary care data set that could be used in a model to predict prostate cancer progression at the time of diagnosis, prior to any treatment being initiated. This approach could allow the identified prognostic factors to be used to develop a new prognostic tool to inform treatment decisions between a patient and their treating team. There are already examples of similar prognostic tools available for use, including Predict Prostate (https://prostate.predict.nhs.uk/). However, these tools have only been developed using secondary care data,[32] which may not capture all important prognostic factors or have equivalent length of follow-up of patients in their development or calibration cohorts. In the context of on-going challenges with prognostication for men with localised prostate cancer and the increasing numbers of men being diagnosed every year, getting the most accurate information to inform treatment discussions between patients and their treating physicians is vital.

### Strengths and limitations

This study has a number of unique features. This is the first study that the authors are aware of to use a primary care data set to identify and confirm prognostic factors associated with prostate cancer progression. CPRD contains all data held in the primary care records of millions of UK patients, allowing the inclusion of a range of potentially important prognostic factors. Using a primary care data set from the National Health Service (NHS) also provided long-term data for included patients, with a mean follow-up of over 14 years. Prolonged follow-up for men with prostate cancer is important as many patients can live for years

before their cancer progresses. The lack of high-quality prognostic research discussed in the introduction is not limited to prostate cancer, with many other prognostic factor studies being conducted in similarly flawed ways.[33–35] This study sought to take a confirmatory approach to postulated prognostic factors in prostate cancer in a rigorous manner, following the methodological recommendations of the REMARK guidelines[36] and the PROGRESS partnership.[14–17]

There are some limitations of this study that need to be considered. Previous research has shown that the prostate cancer registry in England has strong case completeness, but significant missing TNM stage and Gleason score data up until recent years.[23] Data completeness and quality within NCRAS continues to improve, and there is no equivalent UK cancer registry data set with more complete data available at this present time.[22] This level of missing data meant that it was unknown whether the majority of potentially included men had localised disease or not. Even so, the study was still powered to answer the research question, and sensitivity analyses showed minimal changes to almost all relationships between the prognostic factors of interest and the study outcomes. Misattribution of prostate cancer as the primary cause of death may occur in some frail, elderly patients or patients with multimorbidity, affecting the primary outcome of this study. There is evidence of misattribution of prostate cancer as a cause of death in other high-income countries;[37 38] however, an English study comparing death certification to a blinded; independent panel showed that ONS data on prostate cancer mortality classification are highly accurate.[39] This study uses a retrospective design interrogating electronic primary care records. It relies on accurate coding from GPs,[40] and there was significant missing data for some prognostic factors.

This study took a confirmatory approach to identify which prognostic factors for prostate cancer progression may be relevant, and some new prognostic factors not currently recommended for use in clinical practice were identified. These prognostic factors could be used to generate a more robust clinical risk prediction tool to guide treatment decision-making. Developing an accurate prediction tool for prostate cancer progression, not just mortality, could be more useful for informing management discussions between patients and clinicians.

**Contributors** SWDM conceived and designed the work that has led to this submission. He acquired the data and performed the analysis. He drafted the manuscript and approves the final version. He agrees to be accountable for all aspects of the work. As corresponding author, he also confirms he has full access to the data in the study and has taken final responsibility for the decision to submit for publication. SMI played an important role in the data analysis and interpretation of the results. She revised the manuscript and approved the final version. She agrees to be accountable for all aspects of the work. MTM helped design the work that has led to this submission, and supported interpretation of the results. She also

provided study supervision to SWDM. She has revised the manuscript and approved the final version. She agrees to be accountable for all aspects of the work. RMM helped to conceive and design the work that has led to this submission. He also provided study supervision to SWDM. He has revised the manuscript and approved the final version. He agrees to be accountable for all aspects of the work.

**Funding** SWDM is supported by the Can Test Collaborative, which is funded by CRUK (C8640/A23385). This work was supported by an Academic Clinical Fellowship in Primary Care for SWDM, funded by the National Institute for Health Research and Health Education England. The views expressed in this publication are those of the author(s) and not necessarily those of the NHS, the National Institute for Health Research, Health Education England or the Department of Health. RMM was supported by a CRUK programme grant, the Integrative Cancer Epidemiology Programme (C18281/A19169). MM was supported by the NIHR Biomedical Research Centre at University Hospitals Bristol NHS Foundation Trust and the University of Bristol. The funders had no direct role in the planning or undertaking of this study, or the preparation of this manuscript.

**Competing interests** None declared.

**Patient consent for publication** Not required.

**Ethics approval** This study received ethical approval from the Independent Scientific Advisory Committee (ISAC) of the Medicines and Healthcare products Regulatory Authority (MHRA)—Protocol reference 17_041. It was conducted in accordance with the Declaration of Helsinki.

**Provenance and peer review** Not commissioned; externally peer reviewed.

**Data availability statement** Data may be obtained from a third party and are not publicly available. This study analysed a CPRD dataset, with linked NCRAS and ONS data. Permission was not sought to share the dataset publicly.

**ORCID iD**
Samuel William David Merriel http://orcid.org/0000-0003-2919-9087

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
