## [Reviewer comments · BMJ Open]

ARTICLE DETAILS

TITLE (PROVISIONAL)	Retrospective cohort study evaluating clinical, biochemical and pharmacological prognostic factors for prostate cancer progression using primary care data
AUTHORS	Merriel, Samuel; Ingle, Suzanne; May, Margaret; Martin, Richard

VERSION 1 – REVIEW

REVIEWER	Michael Goodman Emory University, USA
REVIEW RETURNED	26-Sep-2020

GENERAL COMMENTS	This is a well-conducted retrospective cohort study described in a well-written manuscript. I have only one substantive concern, which I outline in the following five related comments. 1. It is not entirely clear how the authors distinguished prostate cancer-specific from all-cause mortality. Presumably, this information is available from death certificates and is then recorded in the ONS database. As these are the main endpoints, some description of outcome ascertainment would be helpful. For example, it would be important to know if the data capture immediate, underlying and contributing causes of death, or just the immediate cause of death.2. Some of the results raise concerns about misclassification of the cause of death. As the authors I am sure know, it is not uncommon for prostate cancer patients to die with rather than from the disease. How accurately this information is recorded in a death certificate is unclear. See for example a paper by Moghanaki et al. (Prostate Cancer Prostatic Dis. 2019) that examined this issue in the US, and a paper by Löffeler et al. (Scand J Urol. 2018) evaluating the same research question in Norway.3. I would encourage the authors to address this issue in their data, if possible. I don't know the structure of the ONS data, but if the information on immediate, underlying and contributing causes of death is available, I would consider a sensitivity analysis of some kind. For example, one option would be to restrict the analyses for prostate-cancer specific mortality to only those persons for whom prostate cancer was the only cause of death.4. If a sensitivity analysis is not possible, I would consider a simulation analysis evaluating the possible impact of the cause-of-death misclassification on study results.5. At the very least, this issue should be discussed in the Limitations section.
--

VERSION 1 – AUTHOR RESPONSE

Reviewer: 1

Dr. Michael Goodman, Emory University Rollins School of Public Health

Comments to the Author:

This is a well-conducted retrospective cohort study described in a well-written manuscript.

Thank you for taking the time to review this paper

I have only one substantive concern, which I outline in the following five related comments.

1. It is not entirely clear how the authors distinguished prostate cancer-specific from all-cause mortality. Presumably, this information is available from death certificates and is then recorded in the ONS database. As these are the main endpoints, some description of outcome ascertainment would be helpful. For example, it would be important to know if the data capture immediate, underlying and contributing causes of death, or just the immediate cause of death.

You are correct that the cause of death information comes from death certification data held by the ONS. Only the immediate (or primary) cause of death was made available. The Methods and Materials section has been updated to make this clearer.

2. Some of the results raise concerns about misclassification of the cause of death. As the authors I am sure know, it is not uncommon for prostate cancer patients to die with rather than from the disease. How accurately this information is recorded in a death certificate is unclear. See for example a paper by Moghanaki et al. (Prostate Cancer Prostatic Dis. 2019) that examined this issue in the US, and a paper by Löffeler et al. (Scand J Urol. 2018) evaluating the same research question in Norway.

3. I would encourage the authors to address this issue in their data, if possible. I don't know the structure of the ONS data, but if the information on immediate, underlying and contributing causes of death is available, I would consider a sensitivity analysis of some kind. For example, one option would be to restrict the analyses for prostate-cancer specific mortality to only those persons for whom prostate cancer was the only cause of death.

4. If a sensitivity analysis is not possible, I would consider a simulation analysis evaluating the possible impact of the cause-of-death misclassification on study results.

5. At the very least, this issue should be discussed in the Limitations section.

We recognise the potential issue of death misclassification in prostate cancer patients, and how it could have affected the primary outcome of this analysis. The senior author on this paper was the lead investigator for the CAP PSA trial, which used a blinded independent panel to verify cause of death of trial participants. This trial data was used to compare with English death certification data, and found that ONS death certification in patients with prostate cancer deaths was accurate (See paper by Turner et al in the BJC <https://www.nature.com/articles/bjc2016162>)

As mentioned above, we were not provided with any further details of cause of death other than immediate/primary cause, so further sensitivity analysis is not possible with this dataset. We feel given the recent research using ONS prostate cancer death classification data showed English death certification was accurate, the risk of death misclassification in this patient population is lower. We have added to the Discussion section to highlight this potential limitation and references related to this as per your suggestion.

VERSION 2 – REVIEW

REVIEWER	Michael Goodman Emory University Rollins School of Public Health
REVIEW RETURNED	02-Feb-2021
GENERAL COMMENTS	I have no further comments.